# UNSUPERVISED SIGN LANGUAGE TRANSLATION AND GENERATION

## ABSTRACT

Sign language translation and generation are crucial in facilitating communication between the deaf and hearing communities. However, the scarcity of parallel sign language video-to-text data poses a considerable challenge to developing effective sign language translation and generation systems. Motivated by the success of unsupervised neural machine translation (UNMT), this paper introduces an unsupervised sign language translation and generation network (USLNet), which learns from abundant single-modality (text and video) data without parallel sign language data. Inspired by UNMT, USLNet comprises two main components: single-modality reconstructing modules (text and video) that rebuild the input from its noisy version in the same modality and cross-modality back-translation modules (text-video-text and video-text-video) that reconstruct the input from its noisy version in the different modality using back-translation procedure. Unlike the single-modality back-translation procedure in text-based UNMT, USLNet faces the cross-modality discrepancy in feature representation, in which the length and the feature dimension mismatch between text and video sequences. To address the issues, we propose a sliding window method to align variable-length text with video sequences. To our knowledge, USLNet is the first unsupervised sign language translation and generation model capable of generating both natural language text and sign language video in a unified manner. Experimental results on the BBC-Oxford Sign Language datasets (BOBSL) reveal that USLNet achieves competitive results compared to supervised baseline models, indicating its effectiveness in sign language translation and generation.

## 1 INTRODUCTION

Sign language translation and generation (SLTG) have emerged as essential tasks in facilitating communication between the deaf and hearing communities (Angelova et al., 2022b). Sign language translation involves the conversion of sign language videos into natural language, while sign language generation involves the generation of sign language videos from natural language.

Sign language translation and generation (SLTG) have achieved great progress in recent years. However, training an SLTG model requires a large parallel video-text corpus, which is known to be ineffective when the training data is insufficient (Müller et al., 2022b). Furthermore, manual and professional sign language annotations are expensive and time-consuming. Inspired by the successes of unsupervised machine translation (UNMT) (Artetxe et al., 2018; Lample et al.) and unsupervised image-to-image translation (Liu et al., 2017), we propose an unsupervised model for SLTG that does not rely on any parallel video-text corpus.

In this work, we propose an unsupervised SLTG network (USLNet), which learns from abundant single-modal (text and video) data without requiring any parallel sign language data. Similar to UNMT, USLNet consists the following components: the text reconstruction module (Section 2.1) and the sign video reconstruction module (Section 2.2) that rebuild the input from its noisy version in the same modality, and cross-modality back-translation modules (Section 2.3) that reconstruct the input from its noisy version in the different modality using a back-translation procedure.

Unlike the single-modal back-translation in text-based UNMT, USLNet faces the challenge of cross-modal discrepancy. Sign and spoken languages exhibit distinct characteristics in terms of modality, structure, and expression. Sign language relies on visual gestures, facial expressions, and body

movements to convey meaning, while spoken language depends on sequences of phonemes, words, and grammar rules Chen et al. (2022). The cross-modal discrepancy in feature representation presents unique challenges for USLNet.

To address the cross-modal discrepancy in feature representation, a common practice is to use a linear projection to map the representations from the single-modal representation to a shared multi-modal embedding space (Radford et al., 2021). This approach effectively bridges the gap between different feature representations, facilitating seamless integration of information and enhancing the overall performance of models in handling cross-modal translation tasks. In this work, we propose a sliding window method to address the issues of aligning the text with video sequences.

To the best of our knowledge, USLNet is the first unsupervised SLTG model capable of generating both text and sign language video in a unified manner. Experimental results on the BBC-Oxford Sign Language datasets (BOBSL) reveal that USLNet achieves competitive results compared to the supervised baseline model (Albanie et al., 2021) indicating its effectiveness in sign language translation and generation. (Albanie et al., 2021) is a standard transformer encoder-decoder structure and the encoder and decoder comprise two attention layers, each with two heads.

Our contributions are summarized below:

1. USLNet is the first unsupervised model for sign language translation and generation, addressing the challenges of scarce high-quality parallel sign language resources.
2. USLNet serves as a comprehensive and versatile model capable of performing both sign language translation and generation tasks efficiently in a unified manner.
3. USLNet demonstrates competitive performance compared to the previous supervised method on the BOBSL dataset.

## 2   METHODOLOGY

The proposed framework in this study consists of four primary components: a text encoder, a text decoder, a video encoder, and a video decoder. As illustrated in Figure 1, the USLNet framework encompasses four modules: a text reconstruction module, a sign video reconstruction module, a text-video-text back-translation (T2V2T-BT) module, and a video-text-video back-translation (V2T2V-BT) module. The latter two modules are considered cross-modality back-translation modules due to their utilization of the back-translation procedure. In this section, we will first describe each module and then introduce the training procedure.

**Task Definition**   We formally define the setting of unsupervised sign language translation and generation. Specifically, we aim to develop a USLNet that can effectively perform both sign language translation and generation tasks, utilizing the available text corpus $\mathcal{T} = \{\mathbf{t}^i\}_{i=1}^{M}$, and sign language video corpus $\mathcal{V} = \{\mathbf{v}^j\}_{j=1}^{N}$, where $M$ and $N$ are the sizes of the text and video corpus, respectively.

### 2.1   TEXT RECONSTRUCTION MODULE

As shown in Figure 1, the text reconstruction module uses text encoder and text decoder to reconstruct the original text from its corrupted version. Following the implementation by (Song et al., 2019), we employ masked sequence-to-sequence learning to implement the text reconstruction. Specifically, given an input text $\mathbf{t} = (\mathbf{t_1}, \ldots, \mathbf{t_n})$ with $n$ words, we randomly mask out a sentence fragment $\mathbf{t^{u:v}}$ where $0 < u < v < n$ in the input text to construct the prediction sequence. The text encoder ENC-TEXT is utilized to encode the masked sequence $\mathbf{t}^{\backslash \mathbf{u:v}}$, and the text decoder DEC-TEXT is employed to predict the missing parts $\mathbf{t^{u:v}}$. The log-likelihood serves as the optimization objective function:

$$\mathcal{L}_{\text{text}} = \frac{1}{|\mathcal{T}|} \sum_{t \in \mathcal{T}} log P(\mathbf{t^{u:v}} | \mathbf{t}^{\backslash \mathbf{u:v}}) \tag{1}$$

This task facilitates the model's learning of the underlying text structure and semantics while enhancing its capacity to manage noisy or incomplete inputs.

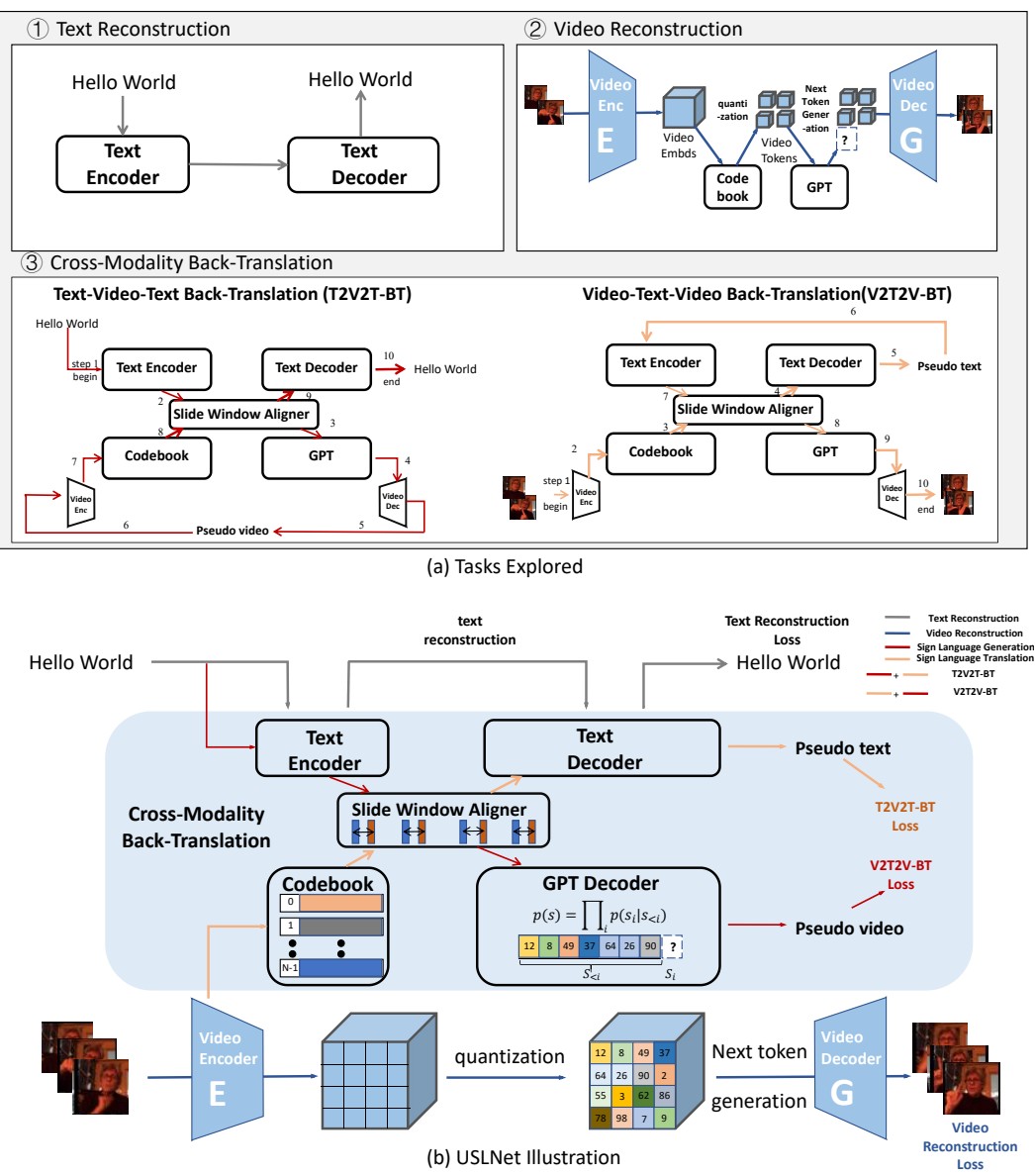

Figure 1: Overview of the proposed USLNet framework and the tasks we explored. USLNet adopts separate encoders to capture modality-specific (visual and textual) characteristics and separate decoders to generate text or video. It employs slide window aligner to archive cross-modality feature transformation. The framework encompasses three modules: a text reconstruction module, a sign video reconstruction module, a text-video-text (T2V2T) module, and a video-text-video (V2T2V) module.

## 2.2 SIGN VIDEO RECONSTRUCTION MODULE

The sign video reconstruction module employs video encoder and video decoder to reconstruct the original video from the downsampled discrete latent representations of raw video data. In this work, we adopt the VideoGPT (Yan et al., 2021) architecture to build the sign video reconstruction module. VideoGPT consists of two sequential stages, i.e., quantization and video sequence generation.

**Quantization** VideoGPT employs 3D convolutions and transposed convolutions along with axial attention for the autoencoder in VQ-VAE, learning a downsampled set of discrete latents from raw pixels of the video frames.

Specifically in the quantization stage, given an input video $\mathbf{v} = (\mathbf{v_1}, \ldots, \mathbf{v_n})$ with $n$ pixels, the video encoder encodes the input $\mathbf{v}$ into video embeddings $\mathbf{E_v} = (\mathbf{E_{v_1}}, \ldots, \mathbf{E_{v_n}})$, then $\mathbf{E_v}$ are discretized by performing a nearest neighbors lookup in a codebook of embeddings $\mathbf{C} = \{\mathbf{e_i}\}_{\mathbf{i=1}}^{\mathbf{N}}$, as shown in Eq.(2). Next, $\mathbf{E_v}$ can be represented as discrete encodings $\mathbf{E_v^q}$ which consists of the nearest embedding indexes in codebook, shown in Eq.(3). Finally, video decoder learns to reconstruct the input $\mathbf{v}$ from the quantized encodings.

$$\mathbf{E_{v_i}} = \mathbf{e_k}, \quad \text{where} \quad \mathbf{k} = \mathrm{argmin_j} \left\| \mathbf{E_{v_i}} - \mathbf{e_j} \right\|_{\mathbf{2}} \tag{2}$$

$$\mathbf{E_v} \to \mathbf{E_v^q} = (\mathbf{k_1}, \ldots, \mathbf{k_n}), \quad \text{where} \quad \mathbf{k_i} = \mathrm{argmin_j} \left\| \mathbf{E_{v_i}} - \mathbf{e_j} \right\|_{\mathbf{2}} \tag{3}$$

The similarity between $\mathbf{E_{v_i}}$ and $\mathbf{e_j}$ serves as the optimization objective function:

$$\mathcal{L}_{\text{codebook}} = \frac{1}{|\mathcal{C}|} \sum_{e_j \in \mathcal{C}} \left\| E_{v_i} - e_j \right\|_2 \tag{4}$$

**Video Sequence Generation** After quantization stage, the discrete video encodings $\mathbf{E_v^q} = (\mathbf{k_1}, \ldots, \mathbf{k_n})$ are feed into the GPT-style decoder, and generate the next video "word" $\mathbf{k_{n+1}}$. The similarity between autoregressively generated video $\mathbf{v_{recon}}$ and the original input video $\mathbf{v}$ serves as the optimization object function:

$$\mathcal{L}_{\text{video}} = \frac{1}{|\mathcal{V}|} \sum_{v \in \mathcal{V}} \left\| v_{recon} - v \right\|_2 \tag{5}$$

## 2.3 CROSS-MODALITY BACK-TRANSLATION MODULE

The cross-modality back-translation module consists of two tasks: text-video-text back-translation (T2V2T-BT) and video-text-video back-translation (V2T2V-BT). In contrast to conventional back-translation (Sennrich et al., 2016), which utilizes the same modality, cross-modal back-translation encounters the challenge of addressing discrepancies between different modalities (Ye et al., 2023). Inspired by the recent work Visual-Language Mapper (Chen et al., 2022), we propose the implementation of a sliding window aligner to facilitate the mapping of cross-modal representations.

**Sliding Window Aligner.** The sliding window aligner is proposed to address the discrepancies between text and video modal representations. Specifically, two primary distinctions between text and video representation sequences are hidden dimensions and sequence length differences. Considering these differences, the aligner consists of two components: *length mapper* $\mathbf{M^L}$ and *dimension mapper* $\mathbf{M^D}$. Considering different back-translation directions (V2T2V and T2V2T), dimension mappers include text-to-video mapper $\mathbf{M_{T \to V}^D}$ and video-to-text mapper $\mathbf{M_{V \to T}^D}$.

Given the text encoder output $\mathbf{E_t}$, the text decoder input $\mathbf{D_t}$, the codebook reconstructed video embedding $\mathbf{E_v}$ and video GPT input $\mathbf{D_v}$, the feature dimension transformation procedure are as follows:

$$\mathbf{D_v} = \mathbf{M^L}(\mathbf{M_{T \to V}^D}(\mathbf{E_t})) \tag{6}$$

$$\mathbf{D_t} = \mathbf{M^L}(\mathbf{M_{V \to T}^D}(\mathbf{E_v})) \tag{7}$$

Sign language constitutes a distinct language system characterized by its unique grammatical principles governing word order. Achieving optimal word order in video-text and text-video tasks poses a significant challenge for Sign Language Translation and Generation (SLTG) models. Furthermore, due to the disparity in decoding termination conditions between video and text modalities, text sequences exhibit variability in length, whereas video sequences maintain a fixed length.

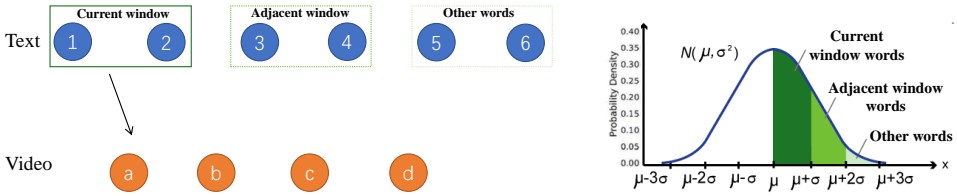

Figure 2: Left: A figure describing slide window aligner at step one. Right: Visualization of the probability distribution (Gaussian distribution) that satisfies the weight coefficients of words in different positions. At step one, we compute the first token of pseudo video "sequence" by slide window aligner.

Aiming to solve the above two challenges, we design **length mapper $\mathbf{M^L}$** method, which uses the sliding window method. According to (Sutton-Spence & Woll, 1999), signing is particularly influenced by English word order when the signers sign while translating from a text. In the context of British Sign Language, presenters may adhere to a more English-like word order and the validation procedure can be seen in Appendix A.1. Drawing upon this linguistic understanding, we propose a method wherein the source sequence is partitioned into distinct windows, allowing each word in the target sequence to align more closely with its corresponding window. Taking text-to-video for example, supposed that input text sequence $\mathbf{t} = (\mathbf{t_1}, \ldots, \mathbf{t_m})$ with m words, video sequence $\mathbf{v} = (\mathbf{v_1}, \ldots, \mathbf{v_n})$ with n frames and $\mathbf{m > n}$, the sliding window method, Length Mapper $\mathbf{M^L}$ which can be described as follows:

$$\mathbf{v_i} = \sum_{i=1}^{n} \alpha_i \mathbf{t_i} \tag{8}$$

$$[\alpha_1 \quad \alpha_2 \quad \ldots \quad \alpha_n] = \text{softmax}\left([\beta_1 \quad \beta_2 \quad \ldots \quad \beta_n]\right),$$

$$\text{specifically} \quad \beta_i \in \begin{cases} (p(\mu + \sigma), \quad p(\mu)], & i \in \text{current window} \\ (p(\mu + 2\sigma), \quad p(\mu + \sigma)], & i \in \text{adjacent window} \\ (p(\mu + 3\sigma), \quad p(\mu + 2\sigma)], & \text{otherwise} \end{cases} \tag{9}$$

Shown in Eq.(8), every video word accept all text words' information. However, each word in the target sequence aligns more closely with its corresponding window. For example, the beginning video frames conveys more information about the first some text words. Specifically, weight coefficient $[\alpha_1, \alpha_2, \ldots, \alpha_n]$ comes from $X = [\beta_1, \beta_2, \ldots, \beta_n]$. X follows a Gaussian distribution $N(\mu, \sigma^2)$. The value of $\beta_i$ depends on where token i is and is divided into three probability intervals $(p(\cdot), p(\cdot)]$, shown in Eq.(9). The value of token $\beta_i$ increases as its proximity to the current window becomes closer. Additionally, the lengths of the window and the stride can be easily calculated using the following rules. The window length is equal to the ratio of the input sequence length to the output sequence length, while the stride is obtained by subtracting the window length from the input length and dividing the result by the output length.

As figure 2 has shown, supposed text has 6 words $\mathbf{t} = (\mathbf{t_1}, \ldots, \mathbf{t_6})$ and video has 4 frames $\mathbf{v} = (\mathbf{v_a}, \mathbf{v_b}, \mathbf{v_c}, \mathbf{v_d})$. We can compute window size = 2, stride = 1. It means the first window is $\mathbf{t_1}, \mathbf{t_2}$, and the corresponding video token is $\mathbf{v_1}$; and the second window is $\mathbf{t_2}, \mathbf{t_3}$ and the corresponding video token is $\mathbf{v_2}$ and so on. When it comes to the first window, $\alpha_1, \alpha_2$ has high probability of Gaussian distribution, $\alpha_3, \alpha_4$ has medium probability of Gaussian distribution, and $\alpha_5, \alpha_6$ has a low probability of Gaussian distribution.

We introduce **dimension mapper $\mathbf{M^D}$** to address the differences in hidden dimensions of different modalities. For example, $\mathbf{M^D_{T \to V}(E_t)}$ transposes text embeddings' hidden dimensions into video embeddings' hidden dimensions, facilitating the integration and alignment of textual and visual information for improved multimodal tasks.

**Cross-Modality Back-Translation.** The T2V2T-BT translates a given text sequence into a sign video, followed by translating the generated sign video back into text, shown in figure 1(a). The objective of T2V2T-BT is to ensure consistency between the generated text and the original text while

accurately translating the video back into the original text. This task assists the model in capturing the semantic and visual correspondence between text and video modalities and comprehending the input data's underlying structure and temporal dynamics.The similarity between back-translated text $\mathbf{t_{BT}}$ and the original input text $\mathbf{t}$ serves as the optimization object function:

$$\mathcal{L}_{\text{T2V2T}} = \frac{1}{|\mathcal{T}|} \sum_{t \in \mathcal{T}} \|t_{BT} - t\|_2 \tag{10}$$

Similarly, the V2T2V-BT task requires the model to translate a given video into its corresponding text description, and then translate the generated text back into a video, using the original video as a reference, shown in figure 1(a). The similarity between back-translated video $\mathbf{v_{BT}}$ and the original input video $\mathbf{v}$ serves as the optimization object function:

$$\mathcal{L}_{\text{V2T2V}} = \frac{1}{|\mathcal{V}|} \sum_{v \in \mathcal{V}} \|v_{BT} - v\|_2 \tag{11}$$

Overall, the cross-modality back-translation module of our proposed USLNet aims to improve the model's ability to translate between text and video modalities in an unsupervised manner, by learning a consistent and meaningful mapping between the two modalities.

## 2.4 Unsupervised Joint Training

The training objective of USLNet combines the above four loss items, which jointly optimize the text and video networks. The loss $L_{text}$ and $L_{video}$ encourages the generator network to generate realistic and diverse texts and videos. while loss $L_{T2V2T}$ and $L_{V2T2V}$ encourage the USLNet to learn a consistent and meaningful mapping between text and video modalities. The objective is to train a model that can generate high-quality sign language videos(texts) from arbitrary text(video) inputs without relying on any labeled data.

$$L_{overall} = \alpha_1 L_{text} + \alpha_2 L_{codebook} + \alpha_3 L_{video} + \alpha_4 L_{T2V2T} + \alpha_5 L_{V2T2V} \tag{12}$$

## 3 Experiment

**Dataset.** The BBC-Oxford British Sign Language Dataset (BOBSL) (Albanie et al., 2021) is a large-scale video collection of British sign language(BSL). The corpus is collected from BBC TV episodes, covering a wide range of topics. It contains 1,004K, 20K, 168K samples in train, dev and test sets, respectively. The vocabulary size is 78K, and out-of-vocab size of the test set is 4.8k.

**Metric.** We adopt the BLEU(Papineni et al., 2002) as the evaluation metric for the sign language translation. For the sign language generation, we follow UNMT (Lample et al.) to utilize back-translation BLEU to assess the performance. Specifically, we back-translate the generated sign language video and use the input text as the reference to compute the BLEU score.

**Model.** The USLNet incorporates the MASS (Song et al., 2019) architecture as the text model backbone and VideoGPT (Yan et al., 2021) as the video model backbone. For the text model, we set the encoder and decoder layers to 6, and the hidden dimension to 1024. As for the video model, we build the VideoGPT with 8 layers and 6 heads,with a hidden dimension of 576. For the codebook, we set it with 2048 codes, wherein each code represents a feature tensor with a 256-dimensional. The training process comprises two stages: pre-training and unsupervised training. Firstly, we perform continued pre-training using the pre-trained MASS model (Song et al., 2019) on the text portion of the BOBSL. Then, we train the VideoGPT model (Yan et al., 2021) on the sign language video component of the BOBSL. Finally, we utilize the pre-trained MASS and VideoGPT models to initialize the USLNet and conduct unsupervised joint training, as described in Section 2.4. We train the whole network for 10 epochs with a learning rate of 1e-3. Moreover, we use greedy decoding in evaluation procedure.

Table 1: Sign language translation performance in terms of BLEU on BOBSL test set. B@1 and denotes BLEU-1 and BLEU-4, respectively.

| Method | Dev | Test | |
|---|---|---|---|
| | B@1↑ | B@1↑ | B@4↑ |
| (Albanie et al., 2021)(supervised) | - | 12.78 | 1.00 |
| (Sincan et al., 2023)(supervised) | 18.80 | 17.71 | 1.27 |
| USLNet(unsupervised) | 17.30 | 21.30 | 0.10 |
| USLNet(supervised) | 19.60 | 15.50 | 1.00 |
| USLNet(unsupervised + supervised Finetune) | 24.60 | 27.00 | 1.40 |

Table 2: Sign language generation performance in terms of back-translation BLEU and Frechet Video Distance (FVD) on BOBSL dataset. B@1 and denotes BLEU-1 and BLEU-4, respectively.

| Method | Dev | | Test | | |
|---|---|---|---|---|---|
| | B@1↑ | FVD↓ | B@1↑ | B@4↑ | FVD↓ |
| USLNet w/o. joint training | 0.50 | 892.8 | 0.70 | 0.00 | 872.7 |
| USLNet w. joint training | 20.90 | 402.8 | 22.70 | 0.20 | 389.2 |

## 4 RESULTS AND DISCUSSION

### 4.1 MAIN RESULT

**Sign Language Translation** We compare the results of USLNet with the supervised approach (Albanie et al., 2021) on BOBSL test set. As illustrated in Table 1, our USLNet achieves competitive results compared to the supervised approach and surpasses it by an 8.0+ BLEU-1 metric. Moreover, unsupervised way can obtain more knowledge representation and is significant for improve supervised translation method (B@4 1.0 –> 1.4). The results show that although translation quality is not perfect, more cases, failure analysis and path to success will be illustrated in Appendix A.2.

**Sign Language Generation** Since there are no existing results for sign language generation on the BOBSL dataset, we compare the use of unsupervised joint training in USLNet. As shown in Table 2, the unsupervised joint training in USLNet yields improvements in terms of back-translation BLEU scores, demonstrating the effectiveness of USLNet. What's more, visual results can be found in Appendix A.6.

### 4.2 ANALYSIS

In this section, to gain a deeper understanding of the improvements achieved by USLNet, we assess the impact of our approach on both the sliding window aligner and our translation outputs.

#### 4.2.1 IMPACT ON SLIDING WINDOW ALIGNER

**Different alignment networks** To further explore the advantages of the proposed sliding window aligner (soft connection), we have designed two comparison aligner networks, altering only the length mapper component $\mathbf{M^L}$. The first network is pooling, where the text sequence is padded to a fixed length and a linear network maps it to the video sequence length. The second network is the sliding window aligner with a hard connection, also utilizing a sliding window mechanism. However, $\alpha_i$ in Eq(9) is non-zero only if tokens are in the current window, indicating that it conveys information exclusively from tokens in the current window.

As demonstrated in Table 3 and Table 4, our method, the sliding window aligner with soft connection, achieves the best performance, exhibiting significant improvements of 9.00 at BLEU-1 score in sign language translation compared to the pooling method and 18.00 BLEU-1 score in sign language generation tasks compared to the pooling method.

Table 3: Sign language translation results of USLNet with different cross-modality mappers on BOBSL. B@1 and denotes BLEU-1 and BLEU-4, respectively.

| Method | Dev | Test | |
| --- | --- | --- | --- |
| | B@1 | B@1 | B@4 |
| Pooling | 10.70 | 12.00 | 0.00 |
| Sliding Window Aligner (hard connection) | 15.50 | 17.10 | 0.00 |
| Sliding Window Aligner (soft connection) | 17.30 | 21.30 | 0.10 |

Table 4: Sign language generation results in terms of back-translation BLEU of USLNet with different cross-modality mappers on BOBSL. B@1 and denotes BLEU-1 and BLEU-4, respectively.

| Method | Dev | Test | |
| --- | --- | --- | --- |
| | B@1 | B@1 | B@4 |
| Pooling | 7.00 | 6.60 | 0.00 |
| Sliding Window Aligner (hard connection) | 11.70 | 11.70 | 0.00 |
| Sliding Window Aligner (soft connection) | 20.90 | 22.70 | 0.20 |

### 4.2.2 IMPACT ON SIGN LANGUAGE TRANSLATION OUTPUTS

The sign language translation performance evaluated by B@4 exhibits considerable room for improvement.To figure out the reasons, we conducted numerous ablation studies and a preliminary analysis utilizing the WMT 2022 sign language translation task, which employed identical challenging data similar to BOBSL.

**Adjusting data distribution benefits SLT.** The transformation of un-parallel video and text data into parallel video and text data, employed in an unsupervised manner, has been demonstrated to significantly improve SLT (+5.60 BLEU-1 socore). This means that we only adjust the training data distribution but do not change how we utilize the data, as we still feed video and text into USLNet in an unsupervised manner. This adjustment, which aligns the video and text modalities, offers notable improvements in the overall performance of unsupervised training. The absence of object offset, which likely contributes to the challenges in utilizing un-parallel data, is a key rationale for this adjustment.

**Freezing Video Encoder has better sign language translation effect** In this research, we compare various freezing strategies by evaluating their impact on the performance of our model, which consists of a text encoder, text decoder, video encoder, and video decoder, inspired by (Zhang et al.). The freezing strategies involve selectively freezing the parameters of different modules during the training process. Specifically, we investigate the effects of freezing the video encoder module while keeping other modules trainable. Our experimental results demonstrate that freezing the video encoder yields superior performance compared to other freezing strategies. This finding suggests that by fixing the video encoder's parameters, the model can effectively leverage the learned visual representations, leading to enhanced feature extraction and improved overall performance in video-related tasks.

**Comparison to WMT sign language translation system** The submissions for the first sign language translation task of WMT 2022 (Müller et al., 2022a) were found to be subpar. These results were similar to the USLNet SLT results achieved by the best submission system MSMUNICH, which obtained a B@4 score of 0.56 on the WMT-SLT test set (all).Several submission reports have highlighted the difficulty of this task, primarily due to the unprecedented scale of the target-side vocabulary, exceeding 20,000 words (Dey et al., 2022). Interestingly, we encountered a similar situation in the BOBSL dataset. Our investigation reveals that BOBSL possesses an extensive vocabulary of 72,000 words, surpassing the 22,000-word vocabulary of WMT, as well as a notable issue of multiple signer bias with 28 signers compared to the 15 signers present in the WMT ALL dataset. These factors serve to illustrate the challenging nature of the BOBSL task. Consequently, the acceptable and promising sign language translation output and evaluation results can be attributed to these factors.

Table 5: Ablation study of USLNet on sign language translation(SLT) on the BOBSL dev set.

| ID | System | SLT B@1$^\uparrow$ |
|:---:|:---:|:---:|
| 1 | Baseline | 3.20 |
| 1.1 | 1+more text data | 9.60 |
| **Explore Multi-task Learning** | | |
| 2.1 | 1.1+ remove text reconstruction at training | 5.40 |
| 2.2 | 1.1+ remove video reconstruction at training | 8.30 |
| 2.3 | 1.1+ remove cross-modality Back-Translation at training | 0.70 |
| **Adjust data distribution** | | |
| 3 | 1.1+ 1M parallel video and text for unsupervised training | 15.20 |
| **Explore Different freezing strategy** | | |
| 4.1 | 3+ freeze video decoder | 10.80 |
| 4.2 | 3+ freeze text encoder | 12.20 |
| 4.3 | 3+ freeze text decoder | 12.60 |
| 4.1 | 3+ freeze video encoder | 17.30 |

## 5 RELATED WORK

**Sign Language Translation** Sign language translation (SLT) focuses on translating sign language video into text (Camgoz et al., 2018). As SLT involves cross-modal learning, previous methods can be broadly categorized into two groups. The first group aims to enrich the visual representation, such as 2D/3D convolution or recurrent encoders (Yin et al., 2021), spatial-temporal multi-cue networks (Zhou et al., 2021b; Yin & Read, 2020), and hierarchical spatial-temporal graph neural networks (Kan et al., 2022). The second group aims to improve the quality of translation output, such as sign language transformers (Camgoz et al., 2020), semi-supervised tokenization approaches (Orbay & Akarun, 2020), and neural machine translation for SLT (Angelova et al., 2022a). Recent studies have investigated MT techniques(He et al., 2022a; 2023) to mitigate data scarcity, such as data augmentation (Ye et al., 2022; 2023; Zhou et al., 2021a) and pretrained language models (Chen et al., 2022). To the best of our knowledge, we are the first to use unsupervised methods for unlabeled data in this SLT domain.

**Sign Language Generation** Sign language generation aims to generate high reliability sign language video (Bragg et al., 2019; Cox et al., 2002). Early efforts relied on classical grammar-based approaches to combine signs for isolated words, achieving continuous sign language production (Glauert et al., 2006b; Cox et al., 2002). Recent work adopt advanced deep learning techniques to generate sign pose (Inan et al., 2022; Saunders et al.). Previous research has predominantly relied on high-quality parallel sign language video and text corpora. We aim to employ an unsupervised approach (Lample et al.; Artetxe et al., 2018; He et al., 2022b) to utilize a large amount of unlabeled data for training SLTG models, which has not been investigated before.

**Text-to-video Aligner and Dual Learning** Due to the inclusion of slide window design as a component of cross-modality, and the similarity between T2V2T-BT and V2T2V-BT to dual learning, we will provide a detailed introduction of text-to-video aligner(Glauert et al., 2006a; Karpouzis et al., 2007; McDonald et al., 2016; Saunders et al., 2020a;b) and dual learning(He et al., 2016; Xia et al., 2017b;a; Yi et al., 2017; Luo et al., 2017) in the Appendix A.3.

## 6 CONCLUSION

In this paper, we present an unsupervised sign language translation and generation network, USLNet, which does not depend on any parallel video-text corpus to perform the cross-modal unsupervised learning task. Experimental results on the BOBSL dataset reveal that USLNet achieves competitive performance compared to the supervised approach, demonstrating its potential for practical applications in sign language translation and generation.

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
