# A APPENDIX

## A.1 VALIDATION BETWEEN SIGN AND TEXT ORDER CONSISTENCY FOR BOBSL

video and glosses are monotonically aligned. However, because BOBSL does not have human-evaluated sentence-level glosses annotations, we suggest that video and text are roughly aligned and align video with text. Morepver, we incorporate validation of BOBSL that video and text are roughly aligned. To address this issue, we must first obtain the golden sign order. In the sign language domain, text-based interpretations of signs are referred to as glosses(Núñez-Marcos et al., 2023). However, BOBSL does not provide sentence-level human-annotated glosses.Therefore, we utilized the automatic gloss annotation released in (Momeni et al., 2022). This gloss annotation consists of word-level annotations, presented as [video name, global time, gloss, source, confidence]. We converted these gloss annotations into sentence-level annotations and assessed the consistency between the gloss (sign) and text orders. From Table 6, we can see the hypothesis that video and text are roughly aligned in BOBSL is right.

Table 6: Validation between sign(gloss) and text order consistency for BOBSL.

|  | Order Consistency |
| --- | --- |
| Strictly Consistent | 0.83 |
| Majority Consistent with two gloss in disorder | 0.87 |
| Main Consistent with three gloss in disorder | 0.91 |

## A.2 QUALITATIVE RESULTS AND FAILURE ANALYSIS

Overall the results in Table 1 and 2 are seemingly very poor. We dig deep into 'why' the results are poor and to work towards building an understanding for "how" they can be improved significantly.

**Regarding the "why" aspect** We conduct a thorough analysis of the results, identifying the areas in which our approach performs well and those that require further improvement.

Initially, we conduct thorough case study including good cases, bad cases and comparison case between USLNet (unsupervised setting ) and (Albanie et al., 2021) which is one supervised model. From digging into our results in Table 7 and 8, we find that we can do relatively better in Main ingredients(eg: bus, I, anything), but always fail in other detail, such as proper noun(eg: Ma Effanga), and complex sentence(which is that).

Table 7: Relatively Good Cases decoded by USLNet in unsupervised settings.

| Good Cases | Case One | Case Two |
| --- | --- | --- |
| Reference | It's quite a journey **especially** if **I get the bus**. | It's hell of a **difference** yeah. |
| USLNet Output | It's **especially** long if I **get the bus**. | It's **different completely**. |

Table 8: Bad Cases decoded by USLNet in unsupervised settings.

| Good Cases | Case One | Case Two |
| --- | --- | --- |
| Reference | Oh, **Ma Effanga** is going to be green. | They started challenging the sultan in a very important aspect, **which is that he is not Muslim enough**. |
| USLNet Output | It's not going to be green. | This is a very important aspect. |

What's more, the comparison case between USLNet (unsupervised setting ) and (Albanie et al., 2021) is as follows.From the table 9 , we observe that our outcomes are competitive with those of supervised methods. Furthermore, in certain instances, we can achieve more accurate output (for example, particularly in specific cases).

Table 9: Comparison Cases between USLNet (unsupervised setting ) and (Albanie et al., 2021).

| Good Cases | Case One | Case Two |
|---|---|---|
| Reference | It's quite a journey **especially** if **I get the bus**. | It's hell of a **difference** yeah. |
| USLNet Output | It's **especially** long if **I get the bus**. | It's **different completely**. |
| Supervised Model Output | How long have you been in the **bus** now. | It was like trying to be **different** to the world. |

**Regarding the "how" aspect**   We propose a two-fold approach. Firstly, we suggest allowing unsupervised learning to serve as a representation learning stage. From the table 1, we can use unsupervised training way can provide one good representation and is significant for improve supervised translation method (B@4 1.0 –> 1.4).Secondly, we recommend enhancing USLNet by focusing on improvements in both the pretraining and aligner components.

USLNet can be divided into two primary components: the pretraining module (comprising the text pre-training module and the video pre-training module) and the mapper part (slide window aligner). Consequently, the paths to success can be categorized into two aspects. The first aspect involves pre-training, where we can adapt our method using multi-modal models, such as videoLLama (Zhang et al., 2023). The second aspect focuses on designing an effective mapper(Saunders et al., 2020a;b).

## A.3   ADDOTIONAL RELATED WORD

**Text-to-Video Aligner**   Text-to-video aligners in sign language domain can be broadly classified into two main categories. The first category involves the use of animated avatars to generate sign language, relying on a predefined text-sign dictionary that converts text phrases into sign pose sequences (Glauert et al., 2006a; Karpouzis et al., 2007; McDonald et al., 2016). The second category encompasses deep learning approaches applied to text-video mapping. (Saunders et al., 2020a;b) adapt the transformer architecture to the text-video domain and employ a linear embedding layer to map the visual embedding into the corresponding space. Unlike these methods, which can only decode pose images, our Unsupervised Sequence Learning Network (USLNet) is capable of generating videos. We address the length and dimension mismatch issues by utilizing a simple sliding window aligner.

Text-to-video aligners in other domains have also been proposed. (Taylor et al., 2012) introduced a method for automatic redubbing of videos by leveraging the many-to-many mapping of phoneme sequences to lip movements, modeled as dynamic visemes. The Text2Video approach (Zhang et al., 2022) employs a phoneme-to-pose dictionary to generate key poses and high-quality videos from phoneme-poses. This phoneme-pose dictionary can be considered as a token-token mapper. Analogously, USLNet quantizes discrete videos and extracts video tokens, a standard technique in the audio domain (Hsu et al., 2021; Wang et al., 2023; Borsos et al., 2023). Consequently, the sliding window aligner also serves as a token-token aligner. However, unlike the Text2Video method, which performs a lookup action to obtain target tokens, our approach decodes the target token using all source tokens.

**Dual Learning**   (He et al., 2016) propose dual learning to reduce the requirement on labeled data aiming to train English-to-French and French-to-English translators. It regards that French-to-English translation is the dual task to English-to-French translation. Thus, it designs to set up a dual-learning game which two agents , each of whom only understands one language and can evaluate how likely the translated are natural sentences in targeted language and to what to extent the reconstructed are consistent with the original. Moreover, researchers exploit the duality between two tasks in training(Xia et al., 2017b) and inference (Xia et al., 2017a) stage , so as to achieve better performance. Dual learning algorithms have been proposed for different tasks, such as translation(He et al., 2016), sentence analysis(Xia et al., 2018), image-image translation(Yi et al., 2017), image segmentation(Luo et al., 2017). USLNet extend dual learning to sign language realm and design dual cross-modality back-translation to learn sign language translation and generation tasks in one unified way.

## A.4 ADDITIONAL ANALYSIS

**Mass text pretraining method outperform than Mlm method**   In this study, we conduct a comparative analysis of various text pretraining methods to assess their impact on sign language translation task shown in Table 10. Specifically, we focus on comparing the performance of the masked language modeling (MLM) (Kenton & Toutanova, 2019) method and the recently proposed masked sequence-to-sequence (Mass) (Song et al., 2019). Our findings reveal that the MASS method outperforms the MLM method (+1.00 B@1) in terms of enhancing the model's ability to capture semantic relationships and improve the overall quality of the learned representations.

**Multi-task modeling benefits SLT**   Multi-task modeling in sign language translation (SLT) presents significant advantages. The incorporation of multiple tasks, particularly the inclusion of cross-modality back-translation, within the modeling framework allows SLT systems to leverage shared representations and tap into a diverse range of informational sources. Our empirical analysis, as depicted in Table 5, substantiates the meaningful impact of key components on SLT performance. Specifically, our findings demonstrate a substantial decrease in SLT results when text reconstruction is omitted (-3.2 B@1), video reconstruction is absent (-1.3 B@1), or cross-modality back-translation training is neglected (-9 B@1). These observations underscore the crucial role of these components in achieving optimal performance in SLT.

Table 10: Additional Ablation study of UnSLNet on sign language translation(SLT) on the BOBSL dev set.

| ID | System | SLT B@1$^\uparrow$ |
|---|---|---|
| 1 | Baseline | 3.20 |
| 1.1 | 1+more text data | 9.60 |
| Adjust data distribution | | |
| 2 | 1.1+ 1M parallel video and text for unsupervised training | 15.20 |
| Explore Different text pretraining method | | |
| 3.1 | 2+ mlm text pretrain method | 15.20 |
| 3.2 | 2+ mass text pretrain | 16.20 |

## A.5 DISCUSSION ABOUT (ALBANIE ET AL., 2021).

In terms of model architecture, both Albanie 2021 and USLNet employ a standard transformer encoder-decoder structure. In the Albanie method, the encoder and decoder comprise two attention layers, each with two heads. Conversely, USLNet adopts a large model architecture, setting the encoder and decoder layers to six. Regarding methodology, Albanie 2021 utilizes a supervised approach for learning sign language translation. In contrast, USLNet employs an unsupervised method, leveraging an abundant text corpus to learn text generation capabilities and employing video-text-video back-translation to acquire cross-modality skills. Concerning model output, Albanie 2021 has released several qualitative examples. We have compared these with the results from USLNet, which demonstrate that USLNet achieves competitive outcomes in comparison to the supervised method.

## A.6 QUALITATIVE VISUAL RESULTS

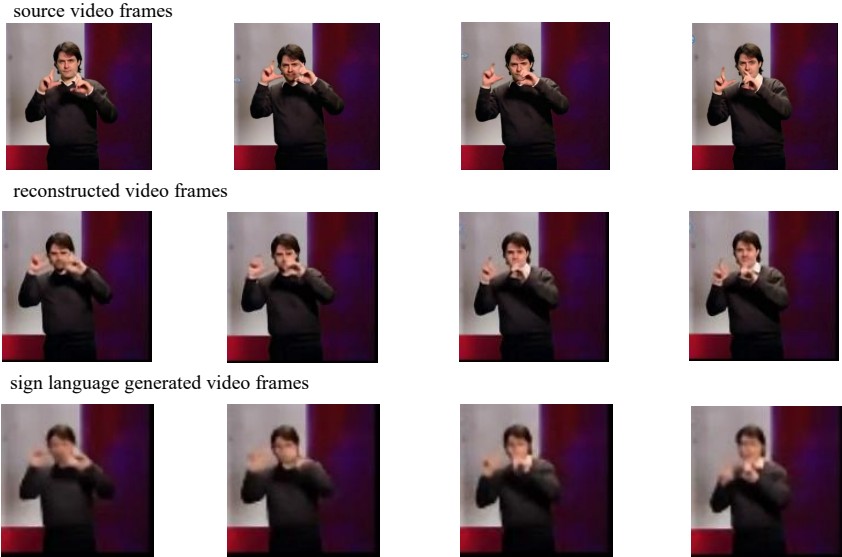

Figure 3: Case study of UnSLNet on BOBSL for sign language generation task. Examples are from test set.

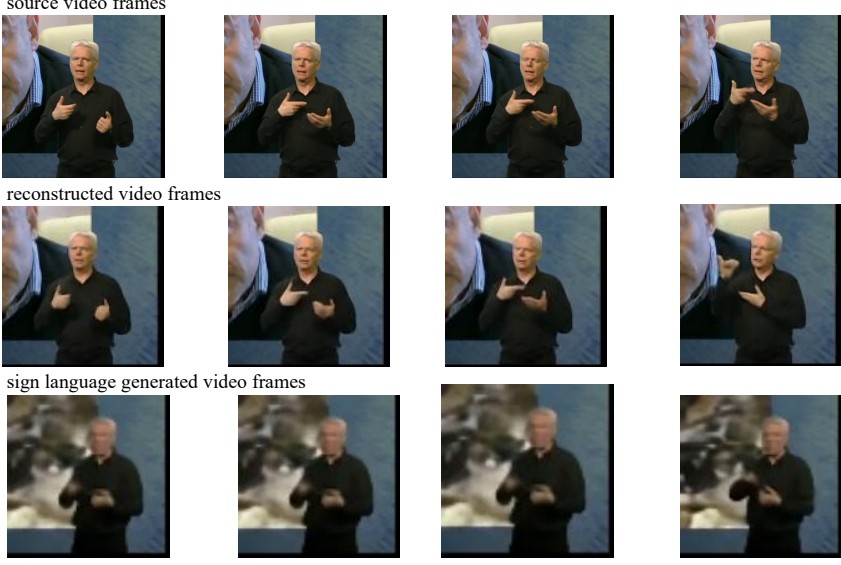

Figure 4: Case study of UnSLNet on BOBSL for sign language generation task. Examples are from test set.

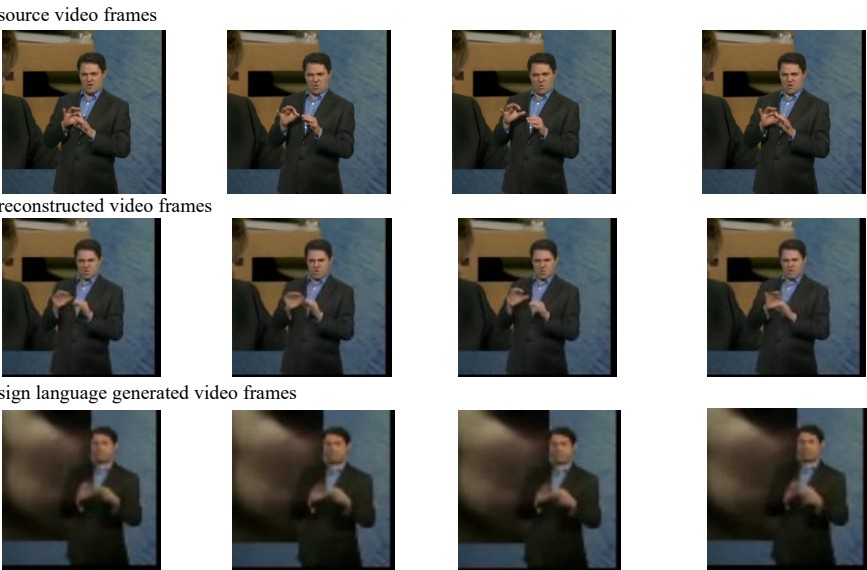

Figure 5: Case study of UnSLNet on BOBSL for sign language generation task. Examples are from test set.