# OpenReview forum: "Unsupervised Sign Language Translation and Generation"
_ICLR.cc/2024/Conference — Submitted to ICLR 2024_

### Official Review · Reviewer_gT8h · 2023-10-28

**Soundness:** 2 fair
**Presentation:** 2 fair
**Contribution:** 2 fair
**Rating:** 5
**Confidence:** 4

**Summary:**

The paper studies a new setting for sign language understanding: unsupervised sign language translation and generation (USLNet), which exploits information from abundant single-modality but non-parallel data. More specifically, UNMT pretrains its text encodes/decoder and video/encoder decoder by reconstruction tasks. To address the misalignment issue between video and texts, the authors further propose a sliding window based aligner.

**Strengths:**

1. The idea is sound. Due to the data scarcity issue in the sign language understanding systems, it is important to explore non-parallel data.
2. The text-video-text and video-text-video back-translation strategies are novel in the sign language community.
3. Detailed ablation studies.

**Weaknesses:**

1. The major issue is the experimental setting.

1.1. Intuitively, leveraging abundant data should be helpful to the model performance. For example, in MMTLB (Chen et al., 2022), using a translation network pretrained on large natural language corpus can boost sign language translation performance. But I didn't see similar conclusions in the experiment section. In Table 1, the authors directly compare a supervised model with the proposed USLNet, and get a worse result on BLEU-4. I understand that the unsupervised performance must be worse, but what is this comparison for? I hope to see that for example, fine-tuning USLNet on parallel corpus can give better results, i.e., similar to the conclusion in MMTLB.

1.2. The performance is **too bad**. Although it may not be the authors' fault (maybe the dataset is too difficult), the poor performance make the comparison less convincing. Experiements on other widely-adopted benchmarks, e.g., Phoenix-2014T and CSL-Daily, shall be considered.

2. It seems that there is a factual error in the sliding window based aligner. The text and video are not monotonically aligned. In fact, only video and glosses are monotonically aligned, e.g., 1-10 frames for the first gloss, and 11-20 frames for the second gloss. Thus, it is questionable for the design of the sliding window-based aligner.

3. The descriptions for the process of the aligner should be more clear. The current form is a bit difficult to understand.

4. The process of two back-translation strategies are simialr to dual learning. The authors may consider adding a subsection in related works to discuss dual learning.

**Questions:**

See weakness.

---

> ### Author Response · Authors · 2023-11-21
>
> Q1：Experimental settings: supervised finetune  results and other dataset results.
>
> 1.1  Fine-tuning USLNet on parallel corpus
> Thanks for your suggestion. We conduct experiments and report the experimental results:
> | **Models**                                     | B@1   | B@4      |
> |------------------------------------------------|-------|----------|
> | **Albanie**                                    | 12.78 | 1.00     |
> | **USLNet(unsupervised )**                      | 21.10 | 0.10     |
> | **USLNet(supervised )**                        | 15.50 | 1.00     |
> | **USLNet(unsupervised + supervised Finetune)** | 27.00 | **1.40** |
>
> The results empirically show that unsupervised training can provide a good representation and significantly improve the supervised translation method (B@4 1.0 --> 1.4).
>
> 1.2 Other dataset results
> For the results on other dataset. Thank you for suggestion. Given that the amount of PHOENIX-2014T and CSL-Daily is too small for training unsupervised model, we do not adopt this results. In the future, we will explore and experiment USLNet on other large sign language dataset, such as OpenASL.
>
>
> Q2： Align problem in slide window aligner.
> Yes, video and glosses are monotonically aligned. However, because BOBSL does not have human-evaluated sentence-level glosses annotations, we suggest that video and text are roughly aligned and align video with text. Morepver, we incorporate validation of BOBSL that video and text are roughly aligned.
>
> To address this issue, we must first obtain the golden sign order. In the sign language domain, text-based interpretations of signs are referred to as glosses[1]. However, BOBSL does not provide sentence-level human-annotated glosses.Therefore, we utilized the automatic gloss annotation released in [2]. This gloss annotation consists of word-level annotations, presented as [video name, global time, gloss, source, confidence]. We converted these gloss annotations into sentence-level annotations and assessed the consistency between the gloss (sign) and text orders.
>
> |                                                | Consistency |
> |----------------------------------------------------|-------------|
> | **Strictly Consistent**                            | 0.83        |
> | **Majority Consistent with two gloss in disorder** | 0.87        |
> | **Main Consistent with three gloss in disorder**   | 0.91        |
>
> We will add the experimental results in the revisd version.
>
> [1] Núñez-Marcos A, Perez-de-Viñaspre O, Labaka G. A survey on Sign Language machine translation[J]. Expert Systems with Applications, 2023, 213: 118993.
> [2] Momeni L, Bull H, Prajwal K R, et al. Automatic dense annotation of large-vocabulary sign language videos[C]//European Conference on Computer Vision. Cham: Springer Nature Switzerland, 2022: 671-690.

---

> ### Author Response · Authors · 2023-11-21
>
> Q3: Descriptions problem for the process of the aligner.
> Thank you for your advice. We will simplify our illustration of the slide window aligner in the revised version. We will add a discussion of dual learning in the related works.
>
> Q4: Dual Learning introduction.
> We appreciate your suggestions regarding dual learning algorithms in our work. Following your recommendation, we will include a subsection discussing dual learning in the revised version.
> Here, we provide a brief outline of the proposed subsection, which will cover the following points:
> Dual Learning. [3] propose dual learning to reduce the requirement on labeled data aiming to train English-to-French and French-to-English translators. It regards that French-to-English translation is the dual task to English-to-French translation. Thus, it designs to set up a dual-learning game which two agents , each of whom only understands one language and can evaluate how likely the translated are natural sentences in targeted language and to what to extent the reconstructed are consistent with the original. Moreover, researchers exploit the duality between two tasks in training[4] and inference [5] stage , so as to achieve better performance. Dual learning algorithms have been proposed for different tasks, such as translation[3], sentence analysis[8], image-image translation[6], image segmentation[7]. USLNet extend dual learning to sign language realm and design dual cross-modality back-translation to learn sign language translation and generation tasks in one unified way.
>
> [3]He D, Xia Y, Qin T, et al. Dual learning for machine translation[J]. Advances in neural information processing systems, 2016, 29.
> [4]Xia, Y., Qin, T., Chen, W., Bian, J., Yu, N., and Liu, T.-Y. Dual supervised learning. ICML, 2017a.
> [5]Xia, Y., Bian, J., Qin, T., Yu, N., and Liu, T.-Y. Dual inference for machine learning. In Proceedings of the Twenty-Sixth International Joint Conference on Artificial Intelligence (IJCAI), pp. 3112–3118, 2017b.
> [6]Yi Z, Zhang H, Tan P, et al. Dualgan: Unsupervised dual learning for image-to-image translation[C]//Proceedings of the IEEE international conference on computer vision. 2017: 2849-2857.
> [7]Luo P, Wang G, Lin L, et al. Deep dual learning for semantic image segmentation[C]//Proceedings of the IEEE international conference on computer vision. 2017: 2718-2726.
> [8]Xia Y, Tan X, Tian F, et al. Model-level dual learning[C]//International Conference on Machine Learning. PMLR, 2018: 5383-5392.

---

> > ### Author Response · Authors · 2023-11-22
> >
> > We thank the reviewer for the review. We have provided clarification in the author's response and also update our paper. Should the reviewer have any further suggestions, we would be happy to provide further clarification and revise our manuscript.

---

> > > ### Comment · Reviewer_gT8h · 2023-11-23
> > >
> > > Thanks for the authors' rebuttal. Most of my concerns are addressed. But I am still a bit disappointed due to the marginal improvement after using the unsupervised model as a pre-trained model. I think the main reason is the dataset is too difficult.
> > >
> > > Indeed, I really like the idea of the paper. I think the authors may keep the idea but think about better evaluation strategy to show the strength of the method. For the current version, I can only keep the borderline rating.

---

### Official Review · Reviewer_zNBk · 2023-10-30

**Soundness:** 3 good
**Presentation:** 3 good
**Contribution:** 2 fair
**Rating:** 5
**Confidence:** 5

**Summary:**

Inspired by the success of unsupervised NMT approaches, this paper proposes USLNet, an unsupervised SL translation and generation approach. USLNet has three main components, namely: text reconstruction module, video reconstruction module and finally cross-modality back translation module. The authors also propose a sliding window based approach to address the alignment issues that are inherent in broadcast SL datasets. The proposed approach is evaluated on BOBSL, however the reported results suggest the proposed approach does not meet the expectation of a translation system (~0.2 BLEU4 score).

**Strengths:**

To the best of my knowledge this is the first bi-directional (translation/generation) SL approach that is trained in an unsupervised manner. Although the results are not promising, the proposed method is sound, and further studying the unsupervised training approach might yield promising results.

**Weaknesses:**

Although I like the idea of using pretrained large-scale models and unsupervised learning, I'd expect quantitative results to back up the benefits of employing these ideas. Sadly, the presented results does not suggest the presented approach to be "working" (~0.2 BLEU-4 score on BOBSL, while the state of the art is above 2 https://openaccess.thecvf.com/content/ICCV2023W/ACVR/papers/Sincan_Is_Context_all_you_Need_Scaling_Neural_Sign_Language_Translation_ICCVW_2023_paper.pdf)

That being said, the reviewers and the readers should acknowledge how challenging the BOBSL dataset is, and that we still need several breakthroughs to progress in large scale SL translation/generation.

Therefore to strengthen the paper, I'd have considered/expected the following:

(1): Experiment on different datasets, such as Phoenix-2014T, or the larger OpenASL and YoutubeASL, which have more state-of-the-art results, hence more data points to gauge the performance/benefits of the proposed approach.

(2): Frame the approach as a pretraining method, and do a final supervised finetuning step (with varying amounts of data). One would expect the unsupervised pretraining on unaligned data to yield better performance than straightforward supervised translation approach, which would have strengthened the utility of the proposed method.

(3) Having some qualitative results and failure analysis for translation/generation would have helped the paper immensely. Relying solely on b1 and b4 results does not give enough insights to the reader, and possibly is not doing the proposed approach justice.

As is, I do not think the reviewer/reader has enough signals to evaluate the benefits of the proposed approach, and I'd highly recommend the authors to consider the suggestions mentioned above.

**Questions:**

(See Weaknesses Section for Suggestions)

--------

After Rebuttal:
I'd like to thank the authors for the rebuttal, additional experiments and considering reviewer's suggestions. As can be seen in their latest experiments, there is benefit to be gained by utilizing the approach as a pretraining step. However, I still would have liked to have more signals from other benchmarks to give the reader better understanding of the proposed approach's performance. Overall I am leaning towards improving my rating to "5: marginally below the acceptance threshold".

---

> ### Author Response · Authors · 2023-11-21
>
> Q1：Other results on BOBSL and acknowledge that the challenging of BOBSL.
> We greatly appreciate your contribution of state-of-the-art (SOTA) results and insights emphasizing the challenges associated with BOBSL. For the work[1], the best model using the context and spotting information achieves 2.88 B@4 points. However, the model can not obtain context and spotting information in our setting. To make a fair comparison, the appropriate baseline for comparison should be B @ 4=1.27 for video-to-text, as depicted in Table 4 in [1], given the fact that USLNet also do not have extra knowledge.
> [1] Sincan O M, Camgoz N C, Bowden R. Is context all you need? Scaling Neural Sign Language Translation to Large Domains of Discourse[C]//Proceedings of the IEEE/CVF International Conference on Computer Vision. 2023: 1955-1965.
>
> Q2：Experiment on different datasets.
> Thank you for suggestion. We follow your suggestion to conduct the experiments and will report the results of Open-ASL in the next few days, given that pretraining and unsupervised training cost lots of time. By the way, because the amount of PHOENIX-2014T is too small for training unsupervised model, we do not adopt this results.
>
> Q3：Unsupervised + Supervised FT.
> Supervised Ft Results: unsupervised way to obtain more knowledge representation:
>
> | **Models**                                     | B@1   | B@4      |
> |------------------------------------------------|-------|----------|
> | **USLNet(unsupervised )**                      | 21.10 | 0.10     |
> | **Albanie(supervised)**                        | 12.78 | 1.00     |
> | **USLNet(supervised )**                        | 15.50 | 1.00     |
> | **Video-text (supervised) [1]**                | 17.71 | 1.27     |
> | **USLNet(unsupervised + supervised Finetune)** | 27.00 | **1.40** |
>
> From the above table, we can use unuspervised training way can provide one good representatio and is significant for improve supervised translation method (B@4 1.0 --> 1.4).

---

> ### Author Response · Authors · 2023-11-21
>
> Q4: Qualitative results and failure analysis.
> Thanks for your suggestions. We completely agree with your opinion that we should dig deep into “why” and “how”. Moreover, it is especially significant for this topic. In the next paragraph, we will give our analysis and possible paths to success.
> 1 **For the reason (“why”) part**, we will divide it into results analysis and dataset analysis. first give the analysis about results to tell which we can do relatively better and show comparison between our output and supervised output.
> 1.1 Case study which can be research on good case, bad case and case for comparison:
> Good cases:
> | **Good Cases**    | Case one                                                  | Case two                            |
> |-------------------|-----------------------------------------------------------|-------------------------------------|
> | **Reference**     | It’s quite a journey **especially** if I **get the bus**. | It’s hell of a **difference** yeah. |
> | **USLNet Output** | It’s **especially** long if I **get the bus**.            | It’s **different completely**.      |
>
>
>
> Bad cases:
> | **Bad Cases**     | Case one                                 | Case two                                                                                                   |
> |-------------------|------------------------------------------|------------------------------------------------------------------------------------------------------------|
> | **Reference**     | Oh, **Ma Effanga** is going to be green. | They started challenging the sultan in a very important aspect, **which is that he is not Muslim enough.** |
> | **USLNet Output** | It’s not going to be green.              | This is a very important aspect.                                                                           |
>
>
>
> From digging into our results, we find that we can do relatively better in Main ingredients(eg: bus, I, anything), but always fail in other detail, such as proper noun(eg: Ma Effanga), and complex sentence(which is that).
>
> Comparison cases:
> |                     | Case one                                                  | Case two                                             |
> |-----------------------------|-----------------------------------------------------------|------------------------------------------------------|
> | **Reference**               | It’s quite a journey **especially** if I **get the bus**. | It’s hell of a **difference** yeah.                  |
> | **USLNet Output**           | It’s **especially** long if I **get the bus**.            | It’s **different completely**.                       |
> | **Supervised Model Output** | How long have you been in the **bus** now.                | It was like trying to be **different** to the world. |
>
> From results comparison, we find that we can find that our results are competitive to supervised method. Moreover, in some ways, we can obtain more accurate output(for example, especially).
>
> 1.2 Dataset Analysis
> As [1] said, the BOBSL dataset is challenging, and that we totally agree with you that it need several breakthroughs to progress in large scale SL translation/generation.
>
> [1]Sincan O M, Camgoz N C, Bowden R. Is context all you need? Scaling Neural Sign Language Translation to Large Domains of Discourse[C]//Proceedings of the IEEE/CVF International Conference on Computer Vision. 2023: 1955-1965.
>
> 2 **For the “how” part**, we propose a two-fold approach. First, we suggest allowing unsupervised learning to serve as a representation learning stage. Second, we recommend enhancing USLNet by focusing on improvements in both the pretraining and aligner components.
>
> 2.1 Supervised Finetuning Results: unsupervised way to obtain more knowledge representation.
>
> | **Models**                                     | B@1   | B@4      |
> |------------------------------------------------|-------|----------|
> | **Albanie**                                    | 12.78 | 1.00     |
> | **USLNet(unsupervised )**                      | 21.10 | 0.10     |
> | **USLNet(supervised )**                        | 15.50 | 1.00     |
> | **USLNet(unsupervised + supervised Finetune)** | 27.00 | **1.40** |
>
> From the above table, we can use unuspervised training way can provide one good representation and is significant for improve supervised translation method (B@4 1.0 --> 1.4).
>
> 2.2 Paths to success:
>
> USLNet can be divided into two primary components: the pretraining module (comprising the text pre-training module and the video pre-training module) and the mapper part (slide window aligner). Consequently, the paths to success can be categorized into two aspects. The first aspect involves pre-training, where we can adapt our method using multi-modal models, such as Video-Llama. The second aspect focuses on designing an effective mapper. We will add the discussion in the revised version.

---

> > ### Author Response · Authors · 2023-11-22
> >
> > We thank the reviewer for the review. We have provided clarification in the author's response and also update our paper. Should the reviewer have any further suggestions, we would be happy to provide further clarification and revise our manuscript.

---

### Official Review · Reviewer_WPdm · 2023-10-30

**Soundness:** 3 good
**Presentation:** 2 fair
**Contribution:** 2 fair
**Rating:** 5
**Confidence:** 4

**Summary:**

This paper develops an approach for unsupervised SL translation and generation entirely using from non-parallel datasets. The motivation is that there is not a lot of paired text and sign language video, so the authors leverage ideas in machine translation and multimodal modeling to build better (unsupervised) sign-text representations.

The approach contains 3 parts: a masked seq2seq text reconstruction module, signing video reconstruction which uses downsampled discrete latent representations (VQ-VAE) with a GPT-style decoder, and back-translation between each modalities to go from text-to-video-to-text and video-to-text-to-video. There is a disconnect in lengths of text and video sequences, so they use a sliding window aligner to map between each.

Results are in some cases better than a supervised baseline on the same dataset and show promise for the approach.

**Strengths:**

* Developing unsupervised approaches for SL generation/translation is important, especially given the many different representations used for signing. One could imagine fine-tuning this approach for any given representation (e.g., Glosses, HamNoSys).
* There are reasonable comparisons to supervised approaches.
* The ablations /sensitivity analysis comparing this approach with different aspects turned off is interesting.
* Given the lack of work in this area, it was valuable to see comparisons such as Table 6 on the WMT 2022 sign language translation task

**Weaknesses:**

Overall the results (e.g., Table 1 & 2) are seemingly very poor. This is by no means a reason to reject a paper, but it does in my opinion require the authors to dig deep into 'why' the results are poor and to work towards building an understanding for how they can be improved significantly. It is nice to see that some results are better than the supervised baseline from Albanie et al., but in an absolute sense they are still low. Are there oracle experiments that could be run? How can the problem be made easier to better understand paths towards success?

One thing that immediately stuck out after going through the appendix is that the visual quality of the SL generations, and likely even the video reconstructions, appear to be too low fidelity to capture important hand or face information. Has there been any experimentation around using different resolution inputs for the video model? Perhaps by doubling or quadrupling the video resolution the model would be able to pick up on more nuance. An alternative approach might be to use key point or whole-body representations (e.g., SMPL) as many recent papers on SL translation have done.

One limitation of the existing approach is that (if I understand correctly) it exclusively trains on BOBSL. On the text encoder side I could imagine it being valuable to leverage existing LLMs and then fine tune. Perhaps the same could be done on the video side? Although I'm not sure what pertaining model or dataset would be the most effective for signings.

On page 5, there is a reference to Sutton-Spence & Woll stating that when signers are translating text then signs will tend to follow the English word order. While this may be true for translating text, it's unclear if it is correct for the datasets used in this paper. Have you validated this on your datasets?

**Questions:**

I would like to see responses to some of the line of inquiry in the weaknesses section.

---

> ### Author Response · Authors · 2023-11-21
>
> Q1: Analysis of the results and possible path to success.
>
> We appreciate your suggestions and agree with your perspective on delving deeper into the "why" and "how" aspects, particularly for this topic. In the following paragraph, we will provide our analysis and potential paths to success.
>
> 1 **Regarding the "why" aspect**, we will conduct a thorough analysis of the results, identifying the areas in which our approach performs well and those that require further improvement.
>
> 1.1 Case study which can be research on good case, bad case and case for comparison:
> Good cases:
>
> | **Good Cases**    | Case one                                                  | Case two                            |
> |-------------------|-----------------------------------------------------------|-------------------------------------|
> | **Reference**     | It’s quite a journey **especially** if I **get the bus**. | It’s hell of a **difference** yeah. |
> | **USLNet Output** | It’s **especially** long if I **get the bus**.            | It’s **different completely**.      |
>
> Bad cases:
>
> | **Bad Cases**     | Case one                                 | Case two                                                                                                   |
> |-------------------|------------------------------------------|------------------------------------------------------------------------------------------------------------|
> | **Reference**     | Oh, **Ma Effanga** is going to be green. | They started challenging the sultan in a very important aspect, **which is that he is not Muslim enough.** |
> | **USLNet Output** | It’s not going to be green.              | This is a very important aspect.                                                                           |
>
>
> From digging into our results, we find that we can do relatively better in Main ingredients(eg: bus, I, anything), but always fail in other detail, such as proper noun(eg: Ma Effanga), and complex sentence(which is that).
>
> Comparison cases:
>
> |                     | Case one                                                  | Case two                                             |
> |-----------------------------|-----------------------------------------------------------|------------------------------------------------------|
> | **Reference**               | It’s quite a journey **especially** if I **get the bus**. | It’s hell of a **difference** yeah.                  |
> | **USLNet Output**           | It’s **especially** long if I **get the bus**.            | It’s **different completely**.                       |
> | **Supervised Model Output** | How long have you been in the **bus** now.                | It was like trying to be **different** to the world. |
>
> From the results, we observe that our outcomes are competitive with those of supervised methods. Furthermore, in certain instances, we can achieve more accurate output (for example, particularly in specific cases).
>
>
> 2 **Regarding the "how" aspect**, we propose a two-fold approach. First, we suggest allowing unsupervised learning to serve as a representation learning stage. Second, we recommend enhancing USLNet by focusing on improvements in both the pretraining and aligner components.
>
> 2.1 Supervised Finetuning Results: unsupervised way to obtain more knowledge representation.
>
> | **Models**                                     | B@1   | B@4      |
> |------------------------------------------------|-------|----------|
> | **Albanie**                                    | 12.78 | 1.00     |
> | **USLNet(unsupervised )**                      | 21.10 | 0.10     |
> | **USLNet(supervised )**                        | 15.50 | 1.00     |
> | **USLNet(unsupervised + supervised Finetune)** | 27.00 | **1.40** |
>
> From the above table, we can use unuspervised training way can provide one good representation and is significant for improve supervised translation method (B@4 1.0 --> 1.4).
>
> 2.2 Paths to success:
>
> USLNet can be divided into two primary components: the pretraining module (comprising the text pre-training module and the video pre-training module) and the mapper part (slide window aligner). Consequently, the paths to success can be categorized into two aspects. The first aspect involves pre-training, where we can adapt our method using multi-modal models, such as Video-Llama. The second aspect focuses on designing an effective mapper. We will add the discussion in the revised version.

---

> ### Author Response · Authors · 2023-11-21
>
> Q2:  Different resolution:
>
> We have conducted experiments using higher resolution; however, the results do not indicate any performance improvement. A possible explanation for this outcome could be that the BOBSL dataset is inherently challenging[1], and increasing the video resolution may further complicate the task. We will add the discussion in the revised version.
>
> [1]Sincan O M, Camgoz N C, Bowden R. Is context all you need? Scaling Neural Sign Language Translation to Large Domains of Discourse[C]//Proceedings of the IEEE/CVF International Conference on Computer Vision. 2023: 1955-1965.
>
> Q3: Pre-training for text and video model:
> In our experiments, we have implemented text and video pretraining to incorporate additional knowledge and utilize it for initialization (Section 2.1 and 2.2). Moreover. Pretraining is indispensable for USLNet.
>
> Q4: Order validation:
> Thank you for your suggestion. Indeed, incorporating validation of BOBSL can make the sliding window aligner more convincing. To address this issue, we must first obtain the golden sign order. In the sign language domain, text-based interpretations of signs are referred to as glosses[2]. However, BOBSL does not provide sentence-level human-annotated glosses.
>
> Therefore, we utilized the automatic gloss annotation released in [3]. This gloss annotation consists of word-level annotations, presented as [video name, global time, gloss, source, confidence]. We converted these gloss annotations into sentence-level annotations and assessed the consistency between the gloss (sign) and text orders.
>
> |                                                | Consistency |
> |----------------------------------------------------|-------------|
> | **Strictly Consistent**                            | 0.83        |
> | **Majority Consistent with two gloss in disorder** | 0.87        |
> | **Main Consistent with three gloss in disorder**   | 0.91        |
>
>
> We will add the experimental results in the revisd version.
>
> [2] Núñez-Marcos A, Perez-de-Viñaspre O, Labaka G. A survey on Sign Language machine translation[J]. Expert Systems with Applications, 2023, 213: 118993.
> [3] Momeni L, Bull H, Prajwal K R, et al. Automatic dense annotation of large-vocabulary sign language videos[C]//European Conference on Computer Vision. Cham: Springer Nature Switzerland, 2022: 671-690.

---

> > ### Author Response · Authors · 2023-11-22
> >
> > We thank the reviewer for the review. We have provided clarification in the author's response and also update our paper. Should the reviewer have any further suggestions, we would be happy to provide further clarification and revise our manuscript.

---

> ### Comment · Reviewer_WPdm · 2023-11-23
> **Keeping ratings**
>
> Thanks for the detailed response. After looking through the details and reading other reviews/rebuttals I am unfortunately keeping my ratings.

---

### Official Review · Reviewer_PET1 · 2023-10-31

**Soundness:** 2 fair
**Presentation:** 3 good
**Contribution:** 2 fair
**Rating:** 5
**Confidence:** 4

**Summary:**

The model is proposed for cross-modal unsupervised learning. It focuses on unsupervised sign language translation and generation and it learns the task without requiring parallel sign language data. The model consists of four modules: text reconstruction,  video reconstruction, text-video-text translation, and video-text-video reconstruction.

**Strengths:**

The overall writing quality is good although there are some issues.

The method is unsupervised which is important in the area as it requires experts to annotate. Also, inspired by unsupervised machine translation and applying the idea to another domain is the originality of the method.

The proposed methods support the writing with detailed formulation and figures.

**Weaknesses:**

Discussion about existing text-to-video aligner algorithms is not sufficient. For example, although text2video[1] is a text-based talking face generation model, it uses an aligner for phoneme-to-pose.

It seems back translations are highly similar to reconstruction loss that is used in image generation, especially in unpaired I2I tasks for cycle consistency. So you might consider elaborating this in the manuscript.

There are no visual results on the manuscript and limited visual results on the supplementary materials. I think it needs to be more convincing that the model is capable of generating sign language videos with high quality.


[1] Zhang, Sibo, Jiahong Yuan, Miao Liao, and Liangjun Zhang. "Text2video: Text-driven talking-head video synthesis with personalized phoneme-pose dictionary." In ICASSP 2022-2022 IEEE International Conference on Acoustics, Speech and Signal Processing (ICASSP), pp. 2659-2663. IEEE, 2022.

**Questions:**

1. The style of equations 10-12 does not fit the manuscript. Authors can consider changing their style to make them consistent with the other equations and the rest of the paper.

2. Why there is no evaluation for the fidelity of the generated videos in terms of well-known metrics such as FID, LPIPS, etc.

3. Why there is no discussion and explanation of the methods proposed in Albanie 2021 in detail as it is the only method that you make a quantitative comparison? I think it needs to be presented more and more importantly the differences and similarities between this and the proposed methods should be highlighted more.

---

> ### Author Response · Authors · 2023-11-21
>
> Q1：Discussion text-to-video aligner algorithms.
> We appreciate your suggestions regarding the text-to-video aligner algorithms in our work. Following your recommendation, we will include a subsection discussing text-to-video aligners in the revised version.
>
> Here, we provide a brief outline of the proposed subsection, which will cover the following points:
> a) Text-to-video aligners in sign language domain can be broadly classified into two main categories. The first category involves the use of animated avatars to generate sign language, relying on a predefined text-sign dictionary that converts text phrases into sign pose sequences [1-3]. The second category encompasses deep learning approaches applied to text-video mapping.  [4-5] adapt the transformer architecture to the text-video domain and employ a linear embedding layer to map the visual embedding into the corresponding space. Unlike these methods, which can only decode pose images, our Unsupervised Sequence Learning Network (USLNet) is capable of generating videos. We address the length and dimension mismatch issues by utilizing a simple sliding window aligner.
>
> b) Text-to-video aligners in other domains have also been proposed.  [6] introduced a method for automatic redubbing of videos by leveraging the many-to-many mapping of phoneme sequences to lip movements, modeled as dynamic visemes. The Text2Video approach [7] employs a phoneme-to-pose dictionary to generate key poses and high-quality videos from phoneme-poses. This phoneme-pose dictionary can be considered as a token-token mapper. Analogously, USLNet quantizes discrete videos and extracts video tokens, a standard technique in the audio domain [8-10]. Consequently, the sliding window aligner also serves as a token-token aligner. However, unlike the Text2Video method, which performs a lookup action to obtain target tokens, our approach decodes the target token using all source tokens.
>
> [1]Glauert, J., Elliott, R., Cox, S., Tryggvason, J., Sheard, M.: VANESSA: A System for Communication between Deaf and Hearing People. Technology and Disability (2006)
> [2]Karpouzis, K., Caridakis, G., Fotinea, S.E., Efthimiou, E.: Educational Resources and Implementation of a Greek Sign Language Synthesis Architecture. Computers & Education (2007)
> [3]McDonald, J., Wolfe, R., Schnepp, J., Hochgesang, J., Jamrozik, D.G., Stumbo, M., Berke, L., Bialek, M., Thomas, F.: Automated Technique for Real-Time Production of Lifelike Animations of American Sign Language. Universal Access in the Information Society (UAIS) (2016)
> [4]Saunders B, Camgoz N C, Bowden R. Progressive transformers for end-to-end sign language production[C]//Computer Vision–ECCV 2020: 16th European Conference, Glasgow, UK, August 23–28, 2020, Proceedings, Part XI 16. Springer International Publishing, 2020: 687-705.
> [5] Saunders B, Camgoz N C, Bowden R. Adversarial training for multi-channel sign language production[J]. arXiv preprint arXiv:2008.12405, 2020.
> [6]Sarah L Taylor, Moshe Mahler, Barry-John Theobald, and Iain Matthews, “Dynamic units of visual speech,” in Proceedings of the 11th ACM SIGGRAPH/Eurographics conference on Computer Animation, 2012, pp. 275–284.
> [7]Zhang, Sibo, Jiahong Yuan, Miao Liao, and Liangjun Zhang. "Text2video: Text-driven talking-head video synthesis with personalized phoneme-pose dictionary." In ICASSP 2022-2022 IEEE International Conference on Acoustics, Speech and Signal Processing (ICASSP), pp. 2659-2663. IEEE, 2022.
> [8]Hsu W N, Bolte B, Tsai Y H H, et al. Hubert: Self-supervised speech representation learning by masked prediction of hidden units[J]. IEEE/ACM Transactions on Audio, Speech, and Language Processing, 2021, 29: 3451-3460.
> [9]Wang C, Chen S, Wu Y, et al. Neural codec language models are zero-shot text to speech synthesizers[J]. arXiv preprint arXiv:2301.02111, 2023.
> [10]Borsos Z, Marinier R, Vincent D, et al. Audiolm: a language modeling approach to audio generation[J]. IEEE/ACM Transactions on Audio, Speech, and Language Processing, 2023.
>
>
> Q2: Similarities and differences between back-translation and reconstruction loss.
> Yes, the loss functions employed in our UNSLT exhibit a high degree of similarity. The distinctions between these two losses stem from the modality of the intermediate representations. In the case of back translation (video-text-video), the intermediate representation is a cross-modal (text) representation, which serves to align the cross-modal representations. Conversely, the intermediate representation for video reconstruction is an unimodal (video) representation, primarily utilized for learning the reconstruction of video sequences.

---

> > ### Author Response · Authors · 2023-11-21
> >
> > Q3: Evaluation for the fidelity of the generated videos and visual results is limited.
> >
> > We appreciate your suggestions regarding the evaluation of video generation. In our study, we have adopted the Frechet Video Distance (FVD) metric [11] to assess generated video quality.
> >
> > The FVD evaluation is conceptually analogous to the Frechet Inception Distance (FID) metric; however, it replaces the feature extraction network for images with a video extraction network, such as the Inflated 3D ConvNet (I3D). The evaluation results are presented below, demonstrating that joint training with USLNet significantly enhances the video generation capability.
> > |                            | FVD ↓ |
> > |----------------------------|-------|
> > | USLNet w/o. joint training | 872.7 |
> > | USLNet w.joint training    | 389.2 |
> >
> > Furthermore, we will include additional visual results in the appendix of the revised version.
> >
> > [11] Unterthiner T, van Steenkiste S, Kurach K, et al. Towards Accurate Generative Models of Video: A New Metric & Challenges[J]. 2018.
> >
> > Q4: Suggestion on equations 10-12.
> > Thank you for your suggestion of the paper writing. We will ensure that the style of the equations is consistent with the rest of the paper, thereby maintaining a coherent and professional presentation throughout the paper.
> >
> > Q5: Discussion about Albanie 2021.
> > Thank you for your suggestions regarding the supervised method. We will include a detailed introduction to the Albanie 2021 method, as outlined below: In terms of model architecture, both Albanie 2021 and USLNet employ a standard transformer encoder-decoder structure. In the Albanie method, the encoder and decoder comprise two attention layers, each with two heads. Conversely, USLNet adopts a large model architecture, setting the encoder and decoder layers to six. Regarding methodology, Albanie 2021 utilizes a supervised approach for learning sign language translation. In contrast, USLNet employs an unsupervised method, leveraging an abundant text corpus to learn text generation capabilities and employing video-text-video back-translation to acquire cross-modality skills. Concerning model output, Albanie 2021 has released several qualitative examples. We have compared these with the results from USLNet, which demonstrate that USLNet achieves competitive outcomes in comparison to the supervised method.
> >
> > |                     | Case one                                                  | Case two                                             |
> > |-----------------------------|-----------------------------------------------------------|------------------------------------------------------|
> > | **Reference**               | It’s quite a journey **especially** if I **get the bus**. | It’s hell of a **difference** yeah.                  |
> > | **USLNet Output**           | It’s **especially** long if I **get the bus**.            | It’s **different completely**.                       |
> > | **Supervised Model Output** | How long have you been in the **bus** now.                | It was like trying to be **different** to the world. |

---

> > > ### Author Response · Authors · 2023-11-22
> > >
> > > We thank the reviewer for the review. We have provided clarification in the author's response and also update our paper. Should the reviewer have any further suggestions, we would be happy to provide further clarification and revise our manuscript.

---

> > > > ### Comment · Reviewer_PET1 · 2023-11-22
> > > >
> > > > Dear Authors,
> > > >
> > > > Thank you for your responses to my comments. I have read the rebuttal and other reviewers comments carefully and I am keeping my score as it is.

---

### Meta-Review · Area_Chair_8g4R · 2023-12-02

**Metareview:**

The paper introduces a model for unsupervised sign language translation and generation, which does not rely on parallel sign language data. The approach includes four modules: text reconstruction, video reconstruction, text-to-video-to-text translation, and video-to-text-to-video reconstruction. USLNet can learn to translate and generate sign language in an unsupervised manner.

The primary concern from the reviewers is that some results reported in the paper are notably poor, raising questions about the model's effectiveness. A deeper analysis of why the results are suboptimal and potential improvements is needed.

The reviewers suggest the paper could explore using the approach as a pretraining method followed by supervised fine-tuning, potentially enhancing the model's performance. The authors tried this approach, and the results show that pretraining is helpful.

The paper primarily focuses on the BOBSL dataset. The reviewers suggest experimenting with other datasets like OpenASL could provide a broader perspective on the model's performance and applicability.

**Justification For Why Not Higher Score:**

There are two major weaknesses:

1. While the paper applies unsupervised methods to a new domain, it's important to note that these techniques have already been extensively explored in other contexts, such as unsupervised machine translation, unsupervised automatic speech recognition, and image style transfer. Consequently, this raises questions about the novelty of the paper's approach.

2. A consistent critique across all reviews is the subpar performance of the results presented. This aspect has been uniformly highlighted as a significant drawback. While it's important to note that a paper's merit isn't solely based on performance, this particular study's reliance on pre-existing ideas makes the effectiveness of the applied approach a critical aspect to evaluate.

**Justification For Why Not Lower Score:**

N/A

---

### Decision · Program_Chairs · 2024-01-16

Reject